

# *In planta* comparative transcriptomics of host-adapted strains of *Ralstonia solanacearum*

Florent Ailloud[1,2], Tiffany M. Lowe[3], Isabelle Robène[1], Stéphane Cruveiller[4,5], Caitilyn Allen[3] and Philippe Prior[1,6]

[1] UMR Peuplements Végétaux et Bioagresseurs en Milieu Tropical, Centre de coopération International en Recherche Agronomique pour le Développement, Saint-Pierre, France
[2] Laboratoire de la Santé des Végétaux, Agence Nationale Sécurité Sanitaire Alimentaire Nationale, Saint-Pierre, France
[3] Department of Plant Pathology, University of Wisconsin-Madison, Madison, WI, United States
[4] Laboratoire d'Analyses Bioinformatiques pour la Genomique et le Metabolisme, Commissariat à l'Energie Atomique et aux Energies Alternatives, Evry, France
[5] UMR CNRS 8030 - Génomique Métabolique, Centre National de la Recherche Scientifique, Evry, France
[6] Département de Santé des Plantes et Environnement, Institut National de la Recherche Agronomique, Sophia Antipolis, France

Corresponding author
Philippe Prior, philippe.prior@cirad.fr

## ABSTRACT

**Background.** *Ralstonia solanacearum* is an economically important plant pathogen with an unusually large host range. The Moko (banana) and NPB (not pathogenic to banana) strain groups are closely related but are adapted to distinct hosts. Previous comparative genomics studies uncovered very few differences that could account for the host range difference between these pathotypes. To better understand the basis of this host specificity, we used RNAseq to profile the transcriptomes of an *R. solanacearum* Moko strain and an NPB strain under *in vitro* and *in planta* conditions.
**Results.** RNAs were sequenced from bacteria grown in rich and minimal media, and from bacteria extracted from mid-stage infected tomato, banana and melon plants. We computed differential expression between each pair of conditions to identify constitutive and host-specific gene expression differences between Moko and NPB. We found that type III secreted effectors were globally up-regulated upon plant cell contact in the NPB strain compared with the Moko strain. Genes encoding siderophore biosynthesis and nitrogen assimilation genes were highly up-regulated in the NPB strain during melon pathogenesis, while denitrification genes were up-regulated in the Moko strain during banana pathogenesis. The relatively lower expression of oxidases and the denitrification pathway during banana pathogenesis suggests that *R. solanacearum* experiences higher oxygen levels in banana pseudostems than in tomato or melon xylem.
**Conclusions.** This study provides the first report of differential gene expression associated with host range variation. Despite minimal genomic divergence, the pathogenesis of Moko and NPB strains is characterized by striking differences in expression of virulence- and metabolism-related genes.

## BACKGROUND

Although the virulence mechanisms of the model *Ralstonia solanacearum* strain GMI1000 have been extensively investigated in model host plants such as *Solanum lycopersicum* (tomato) and *Arabidopsis thaliana*, the genetically diverse *Ralstonia solanacearum* species complex (RSSC) has a cumulative host range that includes more than 250 plant species (*Genin & Denny, 2012*). Very little is known about the prevalence of virulence factors in the RSSC or their contribution to disease across this wide array of hosts. Moreover, recent *in planta* studies have tended to disprove models of virulence regulation that were based on *in vitro* studies (*Jacobs et al., 2012*; *Monteiro et al., 2012*).

RSSC strains cluster in four phylotypes that correspond to strain geographic origin; we focused on phylotype II, which are strains that originated in the Americas. Several groups of strains appear to have adapted to specific hosts; these are often clustered in distinct phylogenetic lineages. The best studied is the brown rot pathotype (phylotype IIB-1), which is a monophyletic group of strains that wilt potatoes in cool highland tropical and temperate zones (*Cellier & Prior, 2010*). Less studied groups include the Moko and not-pathogenic-to-banana (NPB) pathotypes. The Moko pathotype is a polyphyletic group (phylotype IIB3, IIB4, IIA6, IIA24, etc.) of strains that are pathogenic to banana but not to cucurbits (*Cellier & Prior, 2010*). The NPB pathotype is a monophyletic group very close to the IIB4 Moko lineage that has lost the ability to infect banana. However, it is highly virulent on several members of the *Cucurbitaceae* family (*Wicker et al., 2009*; *Wicker et al., 2007*). Remarkably, most of the strains in these two groups are also pathogenic to tomato plants. Thus, the host spectrums of Moko and NPB strains include both exclusive host plants and a common host, tomato. Because NPB and IIB4 Moko strains are genetically similar but have strikingly different biological host ranges, these strains offer a compelling experimental model to pinpoint mechanisms of host specificity in *R. solanacearum*.

We previously used comparative genomic analysis to characterize the gene content and sequence differences between Moko and NPB strains from the IIB4 lineage. The lack of significant genomic differences between these strains, particularly with respect to virulence factors, led us to hypothesize that the different host ranges of Moko and NPB strains can be explained by differential expression of common genes (*Ailloud et al., 2015*).

This work represents the first report of transcriptomic differences associated with RSSC host range variation. It is also the first gene expression analysis of *R. solanacearum* strains from the Moko and NPB pathotypes in the IIB4 lineage. We extracted RNA from Moko strain UW163 (hereafter called the Moko strain) and NPB strain IBSBF1503 (hereafter called the NPB strain) under biologically relevant conditions: minimal medium, rich medium, and during pathogenesis of tomato (both strains), banana (Moko strain) or melon (NPB strain). Differential gene expression was inferred from pairwise comparisons designed to determine either the influence of different environments on individual pathotypes or differences in gene expression between pathotypes (Fig. 1). We observed striking differences between the two pathotypes in levels of type III effector (T3E) genes expression during plant infection, as well as differential regulation of genes encoding siderophore biosynthesis and nitrogen metabolism.

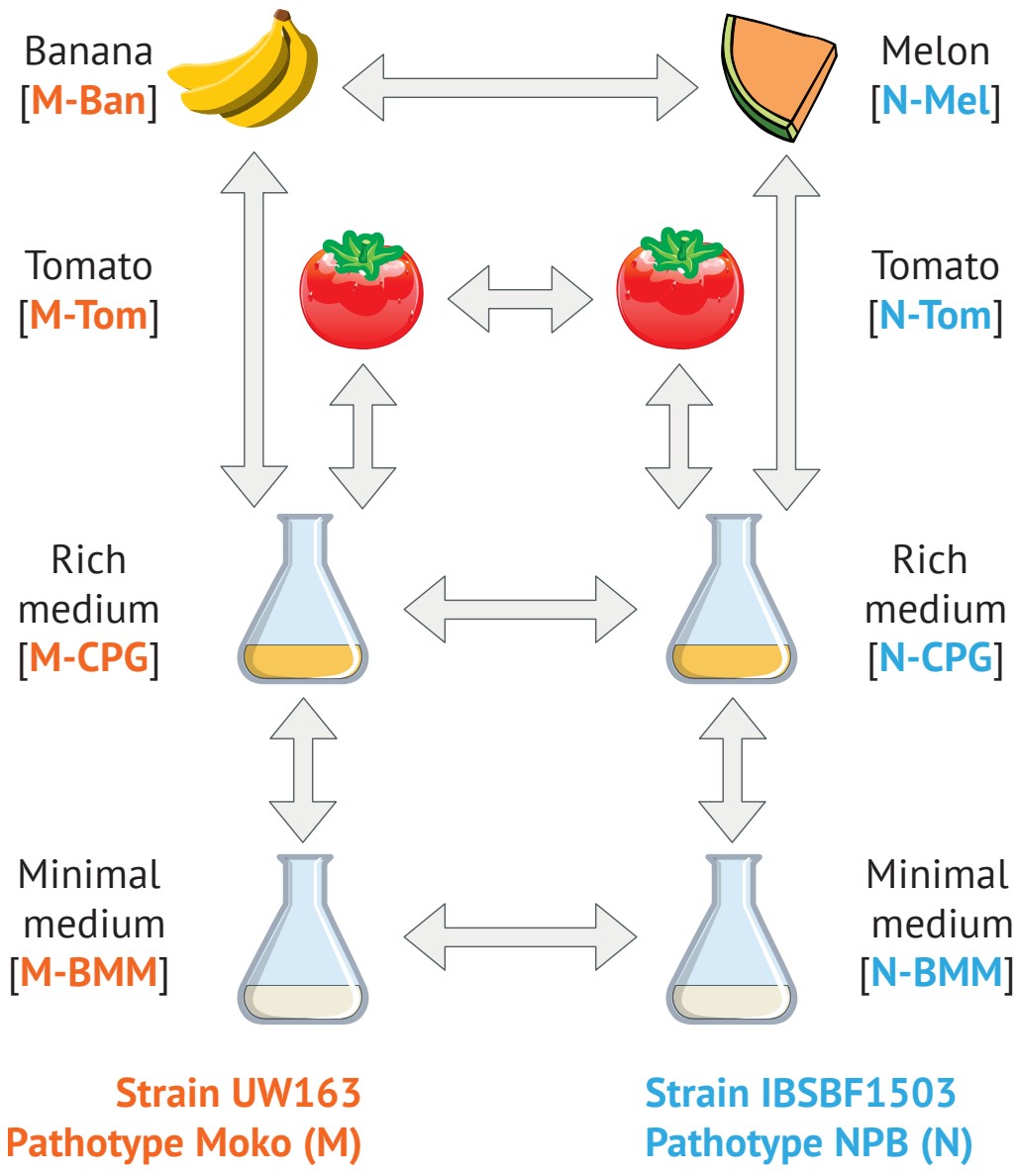

Banana [M-Ban]

Melon [N-Mel]

Tomato [M-Tom]

Tomato [N-Tom]

Rich medium [M-CPG]

Rich medium [N-CPG]

Minimal medium [M-BMM]

Minimal medium [N-BMM]

Strain UW163
Pathotype Moko (M)

Strain IBSBF1503
Pathotype NPB (N)

*R. solanacearum*

**Figure 1 Experimental design.** Each pictogram represents a different condition. Arrows indicate pair of condition that have been analyzed for differential gene expression. M-CPG or N-CPG: Moko (banana) or NPB (not pathogenic to banana) strains grown in rich medium; M-BMM or N-BMM: Moko or NPB strains grown minimal medium; M-Tom or N-Tom: Moko or NPB strain extracted from tomato; M-Ban: Moko strain extracted from banana; N-Mel: NPB strain extracted from melon.

## MATERIALS AND METHODS

### Bacterial strains and growth conditions

*R. solanacearum* strains UW163 and IBSBF1503 were used in this study. UW163 was isolated in 1967 from plantain in Peru; it is classified phylogenetically as phylotype IIB4

and belongs to the banana Moko disease-causing pathotype. IBSBF1503 was isolated in 1999 from cucumber in Brazil; it is classified phylogenetically as phylotype IIB4 and belongs to the NPB pathotype, a pathological variant that is pathogenic on Cucurbits. Both strains were grown aerobically at 28 °C in Boucher's minimal medium (BMM) (*Boucher et al., 1985*) or rich medium composed of casamino acid, peptone and glucose (CPG) supplemented with yeast extract (*Hendrick & Sequeira, 1984*).

### RNA extraction for RNAseq

To observe the transcriptomic landscape of the bacterial cells *in vitro* (i.e., outside plants), the two strains were grown independently in CPG or BMM to a density of $\sim 6 \times 10^8$ CFU/ml (O.D. $= 0.8$). Total bacterial RNA was extracted from bacterial pellets as described (*Jacobs et al., 2012*). Three biological replicates were performed for each condition.

To profile the bacterial transcriptomes during plant colonization, plants were infected with $5 \times 10^8$ CFU via soil-soak inoculation as described in *Cellier & Prior, (2010)* (*Cellier & Prior, 2010*). Tomato plants (wilt-susceptible cv. Bonny Best) were inoculated with either UW163 or IBSBF1503. Banana plants (Cavendish) were inoculated with UW163. Melon plants (cv. Amish) were inoculated with IBSBF1503. Bacteria were extracted from plant stems showing early wilt symptoms at a disease index (D.I.) of 1, which corresponds to wilt symptoms on <25% of leaves. RNA was extracted from bacteria centrifuged out of plant stems as described (*Jacobs et al., 2012*). Only stems containing between $10^8$ and $10^9$ CFU bacteria/g of tissue were used for extraction. Three biological replicates were performed for each condition, and each replicate consisted of a pool of RNA extracted from ~15 stems.

RNA sequencing was carried out by the University of Wisconsin Biotech Center (UWBC, Madison, WI, USA). One hundred base pair single-end libraries were sequenced on an Illumina platform (HiSeq 2000). Each library was sequenced twice to provide technical replicates. Read quality was controlled with FastQC 0.10.1 (www.bioinformatics. bbsrc.ac.uk/projects/fastqc). Low-quality bases and adapters were trimmed using Trimmomatic 0.33 (*Bolger, Lohse & Usadel, 2014*). RNAseq raw data (fastq) are available under BioProject PRJNA297400 for UW163 and BioProject PRJNA297402 for IBSBF1503.

### Genome resequencing

Draft genomes of the *R. solanacearum* strains UW163 and IBSBF1503 were generated in a previous study (*Ailloud et al., 2015*). For this study, we resequenced these strains using long reads generated on a PacBio platform (RS II) at GATC biotech facilities (Konstanz, Germany) and short paired reads generated on an Illumina platform (MiSeq) by UWBC. Long reads were assembled using the SMRT Portal tool (HGAP 3 pipeline) and subsequently corrected with the short paired reads using the PALOMA pipeline (Cruveiller, 2015, unpublished)  with stringent parameters for the "SNP calling" step (allelic frequency was set to 0.8 with at least 10 reads mapping the position and a balance of forward reads to reverse reads was set to 0.33). UW163 BioProject accession number: PRJNA297400; IBSBF1503 BioProject accession number: PRJNA297402. Genomes are also available on the MicroScope platform (www.genoscope.cns.fr/agc/microscope).

## Differential expression analysis

Reads from each library were mapped onto their corresponding reference genome using Bowtie2 2.2.2 (*Langdon, 2015*); the number of uniquely mapped reads for each coding sequence (CDS) was then counted using Bedtools 2.20.1 (*Quinlan, 2014*). Differential gene expression between multiple pairs of conditions was computed using edgeR 3.6.2 (*Robinson, McCarthy & Smyth, 2010*) and DEseq2 1.4.5 (*Love, Huber & Anders, 2014*). Only CDS predicted by both methods with a False Discovery Rate (FDR corrected $P$-value) <0.01 and a $-2 > \log_2$ Fold-Change ($\log_2$ FC) > 2 were considered differentially expressed (DE). When comparing the Moko and NPB strains, we discarded genes duplicated with 100% identity in only one strain. Otherwise, false positive down-regulated genes would be produced in the strain containing two copies when reads mapped to each copy were rejected during the counting phase.

# RESULTS AND DISCUSSION

## Initial results

Mapping indicated that bacterial RNA extraction and library preparation were efficient. For each sample, rRNA and plant RNA contamination were limited, and 60–90% of the total sequenced reads were mapped to CDS, with >10 M mapped reads per sample.

Principal component analysis (PCA) of the counts data revealed that biological and technical replicates were consistent (data not shown). Differentially expressed genes were obtained by merging the results of the edgeR and DEseq2 packages. Although there were some differences between the two sets of results, as expected, these differences were marginal (*Anders et al., 2013*; *Guo et al., 2013*). An inspection of the MA ($\log_2$ FC against log-CPM) and volcano (negative $\log_{10}$ FDR-ajusted $P$-value against $\log_2$ FC) plots suggested no bias between the Count Per Million (CPM) and either the $\log_2$ FC or the FDR-adjusted $P$-value.

Differential gene expression was computed between multiple pairs of samples to identify the differences between the IIB4 Moko and NPB strains *in vitro* and *in planta* during the colonization of several hosts (Fig. 2) (Table S1). Overall, we observed many more differentially expressed genes between bacteria growing in rich CPG and minimal BMM media (~40% of the genome) than in the plant-to-plant comparisons (10–20% of the genome).

To validate our analysis pipeline, we compared the transcriptomic profile of each strain in rich medium (CPG) with the profile in either minimal medium (BMM) or tomato stems. Both the Moko and NPB strains expressed genes encoding the type III secretion system (T3SS) and the secreted T3E much more in BMM and in plants than in CPG. This result is consistent with previous findings in the phylotype I strain GMI1000 that expression of the T3SS and T3E is not repressed in minimal medium or in tomato stems at high cell densities (>$5 \times 10^8$ CFU/ml) in the phylotype I strain GMI1000 (*Jacobs et al., 2012*). We observed similar expression patterns in the samples extracted from banana and melon stems. Together, these results demonstrated that our bioinformatics pipeline functioned as expected in our model and our experimental conditions.

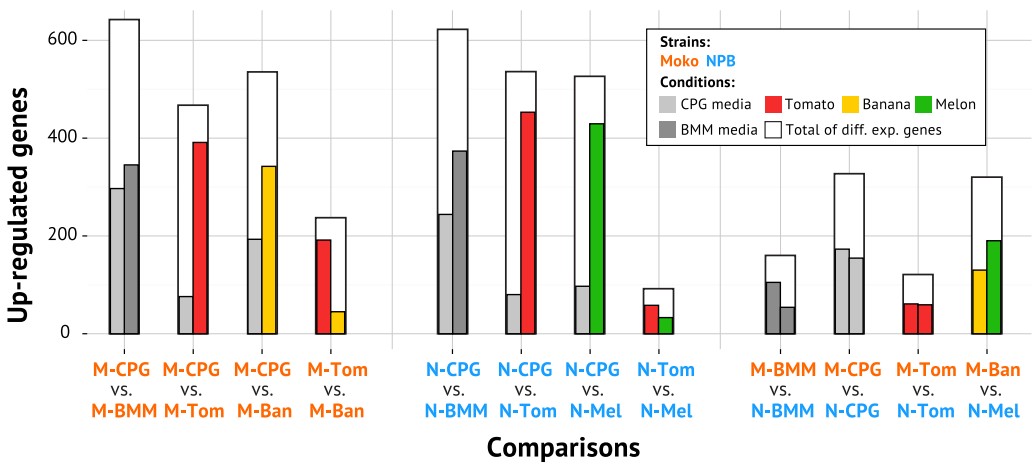

**Figure 2** **Differential gene expression across all tested comparisons.** Each comparison is described by three bars: the first and second inside colored bars corresponds to the number of genes differentially up-regulated in the first and second condition, respectively; the outside white bar indicates the total number of differentially expressed genes in the comparison. M-CPG or N-CPG: Moko (banana) or NPB (not pathogenic to banana) strains grown in rich medium; M-BMM or N-BMM: Moko or NPB strains grown minimal medium; M-Tom or N-Tom: Moko or NPB strain extracted from tomato; M-Ban: Moko strain extracted from banana; N-Mel: NPB strain extracted from melon.

## Conserved regulatory pathways in similar environments: Moko vs. NPB strains in CPG, BMM and the common host plant, tomato

Moko and NPB IIB4 strains share more than 90% of their gene content, and 80% of these common genes are perfectly identical at the nucleotide level (*Ailloud et al., 2015*). By comparing UW163 (Moko) and IBSBF1503 (NPB) gene expression in similar environments (tomato stems, rich CPG and minimal BMM media), we were able to characterize the extent to which these small genomic differences have constitutively modified the transcriptomic profiles of these strains. Detailed results are available in Table S2.

A total of 4,318 genes (90% of the orthologous genes shared by the two strains) were never differentially expressed between the Moko and NPB strains in CPG, BMM and tomato stems based on a $\log_2$ FC threshold of ±2. Conversely, 44 genes (∼1% of the orthologous genes shared by the two strains) were consistently differentially expressed between strains in all similar environments and a majority followed the same regulatory patterns in all conditions tested, suggesting that these genes are differentially expressed constitutively in the two strains. Notably, genes encoding a complete putative amino acid ABC transporter and the catalase KatB were differentially up-regulated in the NPB strain under all three conditions. KatB is required for the pathogen's oxidative stress response during plant infection (*Flores-Cruz & Allen, 2009*). Similarly, a gene encoding a putative hexuronate transporter ExuT2 was differentially up-regulated in the Moko strain in all conditions. Although the role of ExuT2 has not been characterized yet; another hexuronate transporter (ExuT) is involved in the uptake of pectin compounds released during cell wall degradation but it does not contribute directly to wild-type virulence (*Gonzalez & Allen, 2003*). However, ExuT was not differentially expressed in any of the three conditions studied.

An additional 407 genes (9% of the orthologous genes shared by the two strains) had inconsistent expression patterns in the different conditions: i.e., these genes were differentially expressed in one or two of the three conditions. This result could be explained by regulatory pathways that are unique to each strain and that are specifically activated by signals present in only some conditions. Of these genes, 40 were only differentially expressed during tomato infection, 220 only in CPG, and 78 only in BMM. These results could also be due to experimental bias if some parameters were not exactly identical between the Moko and NPB strains in one of the conditions.

It should be noted that mapping transcripts to distinct reference genomes for UW163 and IBSBF1503 can lead to computational bias and false positives. Here, we estimated differentially expressed genes based on homolog families shared by both strains. However, even if homolog families are correctly predicted, a locus can be fragmented or have a slightly different start position in one strain, which allows more reads to be mapped in one locus and eventually gives rise to a false positive.

Most of the genes shared by the Moko and NPB strains were expressed similarly under similar conditions. A total of 10% of the shared genes were differentially expressed, but only 1% of the differences were shared among all three conditions and could be considered constitutively different. Taken together, these results suggest that the Moko and NPB strains use very similar virulence strategies to colonize tomato plants.

## Plant signal-dependent regulation across hosts and pathotypes: growth in rich medium vs. in plants

To pinpoint genes whose expression is modulated by interaction with plants, RNA extracted from each strain during infection of tomato, banana or melon were compared with RNA of the same strain extracted from rich CPG medium. CPG medium was selected as a neutral baseline for comparison with plant samples because BMM medium appears to at least partially mimic the plant environment (*Rahme, Mindrinos & Panopoulos, 1992*; *Tang, Xiao & Zhou, 2006*). Detailed results are given in Table S2.

A total of 1,091 genes were differentially regulated by the plant environment in at least one comparison. These genes were classified into 3 categories in an attempt to describe the influence of each plant-by-strain combination on the pathogens' transcriptomic landscape (Fig. 3).

**Category 1.** A total of 123 genes were differentially expressed in all the plant-by-strain combinations, and the majority of these genes followed a similar pattern of expression. Twenty-four genes exhibited lower expression in plants in both strains. Most of these genes were related to amino acid transport and metabolism according to COG predictions, including the *gcv* (glycine catabolism) and *dpp* (dipeptide uptake) operons. Seventy-seven genes were more highly expressed when the bacteria grew in plants than in rich medium. This group included functions related to energy production or carbohydrate transport and metabolism, including the *cox* (high-affinity cytochrome C oxidase) operon and the *scr* operon. *R. solanacearum* experiences a low-oxygen environment in plant xylem vessels, and a cytochrome C oxidase was shown to contribute to bacterial wilt virulence (*Colburn-Clifford & Allen, 2010*; *Dalsing et al., 2015*). Sucrose uptake and

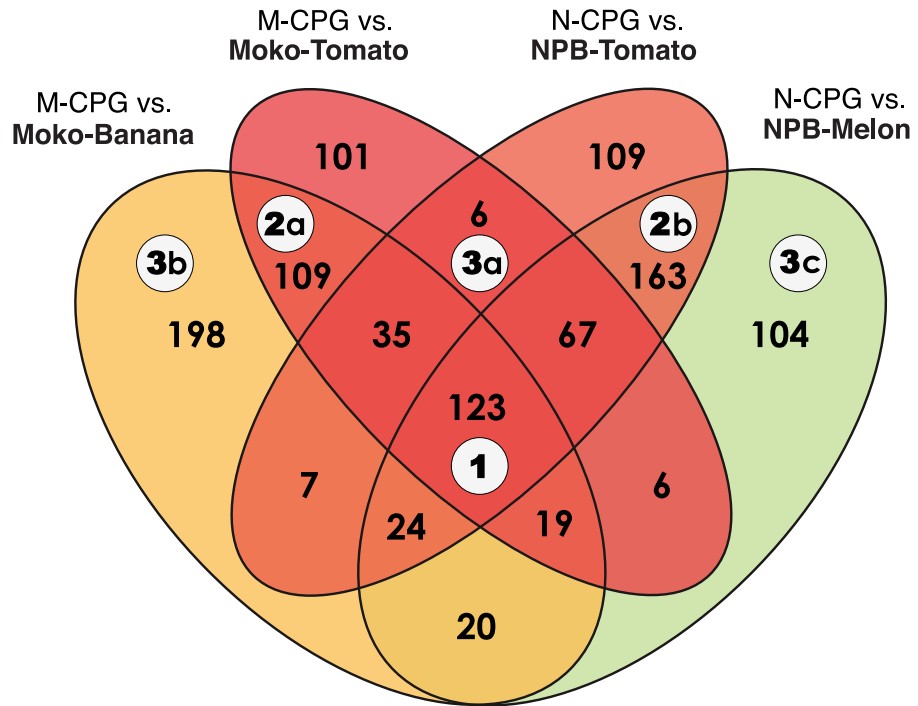

**Figure 3  Venn diagram of differentially expressed genes in *R. solanacearum* strains growing *in planta* compared to in rich medium.**  The numbers in circles correspond to biologically relevant categories. Category 1: genes differentially expressed in all plants compared to rich medium; Category 2: genes differentially expressed in a only a specific strain (Moko (banana): 2a; NPB (not pathogenic to banana): 2b); Category 3: genes differentially expressed in only a specific plant (tomato: 3a; banana: 3b; melon: 3c). M-CPG or N-CPG: Moko or strains grown in rich medium. Moko-Tomato or NPB-Tomato: Moko or NPB strain extracted from tomato; Moko-Banana: Moko strain extracted from banana; NPB-Melon: NPB strain extracted from melon.

catabolism was previously shown to contribute to *R. solanacearum* virulence (*Jacobs et al., 2012*). Importantly, this category also includes 9 of the 69 T3Es shared by these two strains (*ripAM*, *ripAT*, *ripAY*, *ripG3*, *ripR*, *ripC1*, *ripG4*, *ripAN*, and *ripAU*). Four of these T3E are part of the species complex core effectome (*ripAM*, *ripAY*, *ripR* and *ripAN*) (*Ailloud et al., 2015*). Moreover, *ripAN* is one of the few genes (and the only T3E gene) with non-synonymous polymorphisms that are conserved between all sequenced Moko and NPB lineages. Although all these genes are differentially expressed in all conditions and do not diverge across strains and hosts, they provide important information about which genes are modulated by plant signals in these *R. solanacearum* lineages.

A few genes showed divergent expression patterns across strains and plants. The NPB strain differentially up-regulated 18 genes during melon pathogenesis compared with during growth in CPG. These same genes were up-regulated by both strains in tomato plants but were down-regulated in the Moko strain in banana plants. Notable among these 18 genes was the *cco* operon (cytochrome C oxidase cbb-3 type), a high-affinity oxidase that is involved in microaerobic energy metabolism (*Pitcher & Watmough, 2004*) and is required for virulence in tomato plants (*Colburn-Clifford & Allen, 2010*); and the anaerobic ribonucleotide reductase genes *nrdD* and *nrdG*, which are involved

in strictly anaerobic growth. Structural differences leading to different in plant oxygen levels between tomato or melon stems and banana stems, commonly referred to as pseudostems, could explain the Moko-strain-specific expression patterns of these oxygen level-related functions during banana infection.

**Category 2a**. A total of 109 genes were differentially expressed only in the Moko strain in both tomato and banana plants, including 20 Moko-unique genes that are not present in the NPB strain. The gene encoding the endoglucanase *egl*, which degrades plant cell wall and contributes significantly to virulence, was differentially up-regulated *in planta* only in Moko strains, possibly reflecting differences in cell wall structure between monocot and dicot plant hosts. A single T3E, *ripG5*, was up-regulated in the Moko strain in both tomato and banana plants. This effector is part of the core effectome and belongs to the GALA family, which contributes to adaptation to different hosts (*Remigi et al., 2011*).

**Category 2b**. A total of 163 genes were differentially expressed only in the NPB strain in both tomato and melon plants, including 34 NPB-unique not present in the Moko strain. T3SS components, regulators and 34 T3Es were up-regulated in the NPB strains in both tomato and melon plants (*ripZ, ripAC, ripY, ripF1, ripH1, ripG6, ripAO, ripG7, ripD, ripAJ, ripQ, ripAR, ripP1, ripP2, ripE2, ripN, ripL, ripAP, ripAE, ripX, ripAB, ripH2, ripBC, ripH2_2, ripAX1, ripV2, ripBD, ripAW, ripO1, ripS1, ripG2, ripAS, RS_T3E_Hyp12* and a *popC/ripAC*-like effector). Notably, two of these effectors (*ripV2* and *ripAQ*) are not present in the Moko strain. Because these T3E genes are widespread throughout the RSSC including several other Moko lineages (*Ailloud et al., 2015*), *ripV2* and *ripAO* were apparently lost by the IIB4 Moko lineage after its divergence from the IIB4 NPB lineage. Therefore, the differential expression of these genes is unlikely to contribute directly to the unique host range of the NPB strain.

The differential up-regulation of these 34 T3E genes in the NPB strain represents a major difference in type III system regulation between the melon-adapted and banana-adapted strains. Although experimental error cannot be excluded given the number of parameters involved, these results nevertheless appear to be robust. The T3E genes that were exclusively up-regulated *in planta* for the NPB strains (category 2b) were identified using a stringent $\log_2$ FC threshold of $\pm 2$, which corresponds to a 4-fold change in expression between conditions. In the Moko strain, these T3E genes were also more highly expressed in plants than in CPG medium, albeit with a fold change lower than this threshold. These data indicate a greater degree of plant-induced expression of some T3E genes in the NPB strain, not the absence of their induction in the Moko strain.

**Category 3a**. Surprisingly, there were only 6 genes that were specifically differentially expressed during tomato pathogenesis by both the Moko and NPB strains.

**Category 3b**. A total of 198 genes were differentially expressed exclusively in the Moko strain during the colonization of banana plants. This list of genes did not include any T3E. Several adjacent genes with putative functions related to siderophores (RALW3_v3_mp0932 to RALW3_v3_mp0940) and Fur, the ferric uptake regulator, were all up-regulated in banana plants.

**Category 3c.** In total, 104 genes were differentially expressed only in the NPB strain during the colonization of melon plants. The T3E *ripTPS* (a trehalose-6-phosphate synthase) was down-regulated *in planta*, whereas the T3E *ripI* was up-regulated.

Overall, the differential expression of genes during plant pathogenesis compared to in liquid culture appears to be more strongly associated with strains than with hosts. These results suggest that differences in host range between the Moko and NPB strains are not due to specific adaptations to a given host but rather a reshaping of the expression profile to optimize compatibility with different hosts. Among the numerous known virulence factors of *R. solanacearum*, expression of T3SS and T3E genes appears to be differentially triggered by the plant environment in the Moko and NPB strains. Recent studies show that the established model of Type III secretion regulation, shaped by experiments in culture, does not accurately describe the bacterium's behavior inside plants (*Jacobs et al., 2012*; *Monteiro et al., 2012*; *Zuluaga, Puigvert & Valls, 2013*). These recent studies indicate that Type III secretion is robustly expressed *in planta* even after the pathogen has exceeded the cell density that triggers T3SS repression in culture. It is therefore difficult to draw definitive conclusions as to the functional consequences of these results, particularly considering that our data come from strains that are poorly characterized relative to the model strain GMI1000.

## Host-specific differential gene expression: plant vs. plant comparisons

We compared gene expression across strains during pathogenesis of a shared host (tomato), the unique Moko host (banana) and the unique NPB host (melon). To assess intra-strain variations, we compared the tomato and banana transcriptomes of the Moko strain and the tomato and melon transcriptomes of the NPB tomato strain. For the Moko strain, 237 genes were differentially expressed between tomato and banana. Fewer than half as many genes (92) were differentially expressed in the NPB strain between tomato and melon samples. Next, to assess inter-strain variation, the two strains' tomato transcriptomes were compared with one another, and the transcriptome of the Moko strain in banana was compared with that of the NPB strain in melon. The Moko and NPB strains differentially expressed 121 orthologous genes during tomato pathogenesis. When comparing banana and melon, 320 genes were found to be differentially expressed between the Moko and NPB strains. These results were then cross-referenced to identify candidate genes for host specificity (Table 1).

As expected, some genes that were previously identified as being differentially expressed constitutively between the Moko and NPB strains under similar growth conditions (CPG, BMM and tomato plants) were also congruently expressed when these strains were compared in banana and melon plants. The hexuronate transporter gene *exuT2* was up-regulated in the Moko strain in banana plants compared with the NPB strain in melon plants. In the NPB strain, *exuT2* was also down-regulated in melon plants compared with tomato plants. These patterns suggest that ExuT2 is beneficial to *R. solanacearum* fitness in banana but is detrimental or unhelpful in melon. Similarly, the KatB catalase gene was differentially up-regulated by NPB in melon plants compared with its expression level in Moko in banana plants. However, this gene was not down-regulated in the Moko strain

Ailloud et al. (2016), *PeerJ*, DOI 10.7717/peerj.1549

**Table 1** **Candidate genes for host-specific differential gene expression in of *R. solanacearum* strains UW163 (Moko) and IBSBF1503 (NPB).** Log$_2$FC values are indicated for each plant vs. plant comparison (condition 1 vs. condition 2). Bold-faced numbers indicate values within our $-2 > \log_2 FC > 2$ threshold for differentially expressed genes. Genes with negative and positive Log$_2$FC are up-regulated in conditions 1 and conditions 2, respectively. M-Tom or N-Tom: Moko or NPB strain extracted from tomato; M-Ban: Moko strain extracted from banana; N-Mel: NPB strain extracted from melon.

|  | Gene | Product | M-Tom. vs. M-Ban. (log$_2$ FC) | N-Tom. vs. N-Mel. (log$_2$ FC) | M-Tom. vs. N-Tom. (log$_2$ FC) | M-Ban. vs. N-Mel. (log$_2$ FC) |
|---|---|---|---|---|---|---|
|  | *exuT2* | Putative hexuronate transporter | 0.45 | **−2.38** | **−3.72** | **−6.57** |
|  | *katB* | Catalase | −0.97 | 0.46 | **4.63** | **6.05** |
| Iron acquisition | *fur* | Ferric uptake regulator | **2.75** | 0.00 | 0.00 | **−2.07** |
|  | *fyuA* | TonB-dependent siderophore receptor | **3.80** | 0.00 | 0.00 | **−3.83** |
|  | *irp* | Siderophore biosynthesis (NRPS domains) | **4.20** | 0.00 | −0.51 | **−4.14** |
|  | *pchG* | Siderophore biosynthesis (thiazolinyl imide reductase) | **4.67** | 0.00 | 0.00 | **−4.39** |
|  | *angT* | Putative anguibactin biosynthesis (thioesterase) | **4.42** | 0.00 | 0.00 | **−4.18** |
| Microaeroby metabolism | *ccoN* | Cytochrome C oxidase cbb3-type | **−3.70** | 0.55 | −1.04 | **3.21** |
|  | *ccoO* | Cytochrome C oxidase cbb3-type | **−6.06** | 0.72 | −1.61 | **5.16** |
|  | *ccoQ* | Cytochrome C oxidase cbb3-type | **−5.67** | 0.87 | **−2.01** | **4.51** |
|  | *ccoP* | Cytochrome C oxidase cbb3-type | **−5.14** | 0.57 | −1.26 | **4.45** |
| Type III effectors | *ripO1* | Type III effector, HopG1 family | **−2.44** | 0.00 | 1.26 | **3.89** |
|  | *ripP1* | Type III effector PopP1 | −0.65 | 0.00 | 1.56 | **2.25** |
|  | *ripP2* | Type III effector PopP2 | −1.04 | −0.43 | **2.45** | **3.05** |
|  | *ripC2* | Type III effector | −0.70 | 0.00 | **3.73** | **4.27** |
|  | *ripY* | Type III effector (ankyrin repeats) | −1.00 | 0.00 | 0.80 | **2.21** |
|  | *ripS4* | Type III effector SKWP4 | **−3.00** | 0.00 | 0.00 | **2.89** |
|  | *ripE2* | Type III effector (Xcc1246-like) | −1.55 | 0.00 | 1.63 | **2.96** |
|  | *ripD* | Type III effector (AvrPphD family) | −0.78 | 0.00 | 1.29 | **2.32** |
|  | *ripX* | Type III effector (harpin) | −1.86 | 0.00 | 0.93 | **2.88** |
|  | *ripAB* | Type III effector | **−2.21** | 0.00 | 1.22 | **3.31** |
|  | *ripAC* | Type III effector (LRR domain) | −1.10 | 0.00 | 1.20 | **2.14** |

**Table 1** (*continued*)

| | Gene | Product | M-Tom. vs. M-Ban. (log$_2$ FC) | N-Tom. vs. N-Mel. (log$_2$ FC) | M-Tom. vs. N-Tom. (log$_2$ FC) | M-Ban. vs. N-Mel. (log$_2$ FC) |
|---|---|---|---|---|---|---|
| Nitrate assimilation | *nirB* | Assimilatory nitrite reductase | 0.00 | 0.00 | −1.67 | **−2.37** |
| | *nirD* | Assimilatory nitrite reductase | 0.00 | 0.00 | **−2.35** | **−2.08** |
| | *narK3* | ATP-independent high-affinity nitrate transporter | 0.00 | −1.02 | −1.10 | **−2.59** |
| | *narK1* | NO$_3^-$ ABC transporter protein | **−6.88** | 0.84 | −1.27 | **6.44** |
| | *narK2* | NO$_3^-$ ABC transporter protein | **−6.43** | 0.98 | −1.37 | **6.05** |
| | *narG* | NO$_3^-$ reductase | **−5.75** | 0.00 | −1.07 | **5.24** |
| | *narH* | NO$_3^-$ reductase | **−5.05** | 0.00 | −1.04 | **4.47** |
| Denitrification | *narI* | NO$_3^-$ reductase | **−4.85** | 0.00 | −1.06 | **4.30** |
| | *narJ* | NO$_3^-$ reductase | **−5.16** | 0.00 | −1.22 | **4.48** |
| | *narX* | NO$_3^-$ sensor kinase | **−4.21** | 0.00 | −0.96 | **3.34** |
| | *narL* | NO$_3^-$ response regulator | **−3.62** | 0.45 | −0.97 | **3.09** |
| | *hmp* | Flavohemoprotein, NO dioxygenase | **−4.87** | 0.72 | −0.58 | **4.99** |
| | *aniA* | NO$_2^-$ reductase | **−7.37** | 0.00 | −1.55 | **6.29** |
| | *norB* | NO reductase | **−5.47** | 0.87 | −0.93 | **5.41** |

in banana plants compared with tomato plants. Although the roles of these two genes in virulence are not clearly defined, they were among the most differentially expressed genes between the Moko and NPB strains under all the conditions tested, with a $\log_2$ FC of up to 6, corresponding to a ~65-fold change in expression.

In the Moko strain, the ferric iron uptake regulator *fur* and other genes related to siderophore biosynthesis were more highly expressed during banana pathogenesis compared with either the Moko strain in tomato or the NPB strain during melon pathogenesis. Considering that these same genes were also up-regulated by the Moko strain exclusively in banana plants compared with rich medium (category 2a), it appears that they are specifically induced by the banana environment and may contribute to bacterial fitness in this host. Iron-scavenging by *R. solanacearum* has been investigated to some degree, and a previous study found that siderophores did not contribute to pathogenicity of North American IIA7 strain AW on tomato (*Bhatt & Denny, 2004*). However, tomato xylem is not iron limiting, and transgenic tomato plants that display iron-binding activity are more resistant to *R. solanacearum*, suggesting that iron availability could be a determining factor for virulence in other hosts (*Bhatt & Denny, 2004*). Moreover, some plant pathogenic *Erwinia* species do require siderophore production for full virulence (*Dellagi et al., 1998*; *Expert, 1999*). Further investigation of the iron content in banana xylem sap, studies of the *in planta* regulation of ferric iron uptake, and determining the virulence of *fur* mutants in Moko and NPB strains, would help determine whether siderophores contribute differentially to virulence in banana plants and thus to the distinct host ranges of these two strains.

On the other hand, the *cco* high-affinity oxidase operon (cytochrome C oxidase cbb-3 type) was down-regulated during banana pathogenesis compared with either the Moko strain in tomato or the NPB strain during melon pathogenesis. This operon was also down-regulated in banana relative to its expression in medium, whereas it was up-regulated in melon for the NPB strain and in tomato for both strains (category 1). This result suggests that oxygen scavenging may be less important for *R. solanacearum* banana pathogenesis than for pathogenesis of either tomato or melon.

In the NPB strain, genes encoding eleven T3Es (RipO1, RipP1, RipP2, RipC2, RipY, RipS4, RipE2, RipD, RipX, RipAB, and RipAC) were differentially up-regulated by NPB in melon plants compared with their expression by the Moko strain in banana plants. *ripC2* was the only effector that was differentially up-regulated by the NPB strain in tomato plants compared with the Moko strain. The expression profiles of *ripP1*, *ripP2*, *ripY*, *ripE2*, *ripD*, *ripX*, *ripAB*, and *ripAC* were as expected, because these genes were also up-regulated exclusively in the NPB strain *in planta* compared with rich medium (category 2b). However, these effectors did not display any intra-strain variation. They were thus induced by the plant environment only in the NPB strain and were not influenced by differences between the tomato and melon environments. RipX, RipAB and RipAC are encoded by a HrpB-activated operon (formerly referred as *popABC*) and are not required for virulence on tomato or for causing the hypersensitive response (HR) on tobacco (*Gueneron et al., 2000*). Moreover, RipAB and RipAC belong to the core effectome. RipP1, a member of the YopJ/AvrBsT family, is an avirulence factor that prevents strain

GMI1000 from causing disease on some *Petunia* cultivars and *Arabidopsis thaliana* (*Lavie et al., 2002*). In the Moko strain, *ripO1*, *ripS4* and *ripAB* were also down-regulated in banana plants compared with tomato plants; their expression in the Moko strain might thus be detrimental to bacterial fitness in banana plants.

Finally, the Moko and NPB strains appeared to favor distinct nitrogen metabolic pathways in banana and melon plants. Two genes involved in nitrate assimilation, the assimilatory nitrite reductase encoded by *nirBD* and a nitrate uptake transporter encoded by *narK3*, were differentially up-regulated by the Moko strain in banana plants compared with their expression by the NPB strain during melon pathogenesis. *R. solanacearum* is known to metabolize nitrate from the xylem sap, and nitrate assimilation contributes to virulence during early-stage tomato infection (*Dalsing & Allen, 2014*). Conversely, the denitrification genes *narHIJGKLX*, *aniA*, *norB*, and *hmp* were differentially up-regulated by the NPB strain in melon plants compared with their expression by the Moko strain during banana pathogenesis. In the Moko strain, these genes were also relatively less expressed in banana plants than in tomato plants. These genes encode the pathway that carries out partial denitrification (step-wise reductive conversion of nitrate to nitrous oxide) and nitric oxide detoxification; these reactions occur under low-oxygen conditions. In GMI1000, both denitrification and nitric oxide detoxification are required for wild-type virulence and growth in tomato xylem (*Dalsing et al., 2015*). Our finding that the denitrification pathway and high-affinity oxidase CcoN1 are both expressed at lower levels in banana than in tomato is further indirect evidence that the Moko strain experiences a lower oxygen environment in tomato xylem than in the xylem of the banana pseudostem. Direct measurement of *in planta* oxygen levels in the two hosts would conclusively test this hypothesis. The relative biological advantages of higher or lower xylem oxygen levels for *R. solanacearum* remain to be determined. Together, these results suggest that the colonization of the banana xylem requires metabolic adaptation by *R. solanacearum*.

## CONCLUSIONS

In contrast with our previous comparative genomics study that found few differences in gene content and sequence between Moko and NPB strains of *R. solanacearum*, this transcriptomic analysis revealed significant differences between these phylogenetically close but biologically divergent plant pathogens. The Moko and NPB strains exhibited much more divergent transcriptomic profiles than might have been predicted by their gene content, especially during bacterial wilt pathogenesis. Gene expression was generally convergent between the two pathotypes in similar environmental conditions, but the expression levels of several virulence-associated genes diverged during infection of their unique hosts. In banana plants, the Moko strain differentially up-regulated siderophore biosynthesis and nitrate assimilation, whereas in melon plants the NPB strain differentially up-regulated expression of certain T3E and denitrification. The differences in gene expression upon plant cell contact were more strain-specific than host-specific, particularly with respect to virulence-related genes. Future functional studies should address the degree to which the differential expression of T3Es, siderophores and nitrogen

metabolic pathways are involved in the distinct host ranges of *R. solanacearum* Moko and NPB strains. Furthermore, this study focused solely on the transcriptomic behavior of *R. solanacearum* after the first wilting symptoms become apparent and the vascular infection is already established. Additional studies are required to characterize the expression profiles of host-adapted strain during root colonization and pre-symptomatic disease stages as well as during incompatible interactions when the Moko strain attempts to infect melon or the NPB strain attempts to infect banana.

### Funding

The European Union (ERDF), Conseil Régional de La Réunion, the French Agence Nationale de la Recherche and CIRAD provided financial support. ANSES secured part of the doctoral fellowship awarded to F. A. (contract Number 11-237/BSL). T.M.L. was funded by NIH National Research Service Award T32 GM07215 and by Agriculture and Food Research Initiative Competitive Grant # 2015-67011-22799 from the USDA National Institute of Food and Agriculture. The funders had no role in study design, data collection and analysis, decision to publish, or preparation of the manuscript.

### Grant Disclosures

The following grant information was disclosed by the authors:
The European Union (ERDF).
Conseil Régional de La Réunion.
French Agence Nationale de la Recherche.
CIRAD.
NIH National Research Service Award: T32 GM07215.
Agriculture and Food Research Initiative Competitive: 2015-67011-22799.

### Competing Interests

The authors declare there are no competing interests.

### Author Contributions

- Florent Ailloud conceived and designed the experiments, performed the experiments, analyzed the data, wrote the paper, prepared figures and/or tables, reviewed drafts of the paper.
- Tiffany M. Lowe performed the experiments, wrote the paper, prepared figures and/or tables, reviewed drafts of the paper.
- Isabelle Robène contributed reagents/materials/analysis tools, reviewed drafts of the paper.
- Stéphane Cruveiller analyzed the data.
- Caitilyn Allen conceived and designed the experiments, wrote the paper, reviewed drafts of the paper.
- Philippe Prior conceived and designed the experiments, performed the experiments, wrote the paper, reviewed drafts of the paper.

## DNA Deposition

The following information was supplied regarding the deposition of DNA sequences:

Genbank PRJNA297400

Genbank PRJNA297402.

## Supplemental Information

Supplemental information for this article can be found online at http://dx.doi.org/10.7717/peerj.1549#supplemental-information.

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
