# Peer review of "In planta comparative transcriptomics of host-adapted strains of Ralstonia solanacearum"

_PeerJ, doi:10.7717/peerj.1549_

## Round 0.1 · original submission · Minor Revisions

Dear Authors, Dear Florent Ailloud et Philippe Prior,

We have received reviews from two well-informed reviewers on the manuscript and I have also read it myself. All comments are congruent in that it is a valid and well-presented study generating a wealth of transcriptomics data of the two Rs species which will be interesting to many researchers in the field.

Please note the comments of the reviewers and integrate them into the discussion / conclusions.

Also make sure, as in particular reviewer 2 points out, to add the project accession numbers for the RNA-seq experiments to the M&M section of the manuscript.

Thanking you very much for sending your fine manuscript to PeerJ.

Kind regards,
Christine Josenhans

Reviewer 1 ·

Basic reporting

In this manuscript, Aillound and coworkers examined the expression profiles of two related strains of the plant pathogen Ralstonia solanacearum called Moko and NPB. A previous study revealed that Moko and NPB are highly similar at the genomic level, however, their host range is quite different (Moko can infect banana, while NBP has lost this ability). To identify, the cause(s) for the different host range, the authors employed RNA-seq to determine the transcriptomes of both strains. Several experimental conditions were tested, including direct contact of the two strains with host plants. Differentially expressed genes were determined, categorized and discussed in the context of their potential role in host-microbe interactions. The manuscript includes a wealth of transcriptomic data which should be interesting to many scientists in the field.Since the manuscript contains no follow-up experiments it remains largely descriptive and, as a result, lacks scientific novelty. However, it does work as an initial screen for genes relevant for pathogenicity of Ralstonia strains.

Experimental design

The study is well motivated, the experimental design is solid and the description in the text is clear.

Validity of the findings

Three independent biological replicates of each condition were performed providing sufficient statistical significance.

Please add the accession number(s) for the RNA-seq experiments to the M&M section of the manuscript.

Additional comments

As indicated above, I think that a few additional experiments focusing on the differentially expressed transcripts identified in this study (e.g. encoding genes involved in siderophore production) could improve the impact of this study.

Reviewer 2 ·

Basic reporting

This manuscript is well-prepared. The structure and overall content are clear. However, a few mistakes should be corrected. For instance (but not limited to these): line 28-30, line 208.

Experimental design

This study aimed to investigate the nature of host specificities between two genetically closely related Rs strains by transcriptomic analysis. Overall, the experimental designs mostly fit the goal of this study. Comprehensive comparative RNA Seq analyses of good-qualities were carried out and massive useful transcriptome data were generated. Methods are sufficiently described. These would certainly benefit the scientific communities.
A few questions/comments: (1) Were the materials collected from three biological repeats for each experimental condition pooled and subjected to RNA Seq, or they were subjected to RNA Seq separately? (2) To establish a successful infection, Rs needs to overcome many hostile factors throughout its way and colonization in plant root and stembase at the early infection stage is very critical, especially when Rs encounters different host plant species. In this study, in planta bacteria were collected from stems of infected plants with an average wilting score of 1.0 and containing a bacterium density higher than 108 CFU/g tissue, when the bacterium has complete its infection by secreting a large amount of exopolysaccharides. Therefore, the results reveal mainly the shared as well as different features of the two strains at the final stage of bacterial pathogenesis, rather than the most critical initial stage of infection. (3) It is a pity that Moko-melon and NPB-banana samples were not included for the comparative transcriptome analyses to make further comprehensive and appropriate comparisons. This is particularly critical for finding and comparing decisive factors regulating host specificities at the initial infection stage between the two Rs strains.

Validity of the findings

The data were well-analyzed and statistically sound. This study provides new information regarding RSSC pathogenesis in the corresponding susceptible host plants. It is recommended that the points I raised above should be taken into consideration for the discussion and conclusions.

---

## Round 0.2 · accepted · Accept

Dear Drs. Caitilyn Allen, Florent Ailloud and Philippe Prior,

Thank you for your thorough revision, which addresses all reviewers' comments sufficiently, and for careful editing of the manuscript.

Best wishes,

Christine Josenhans
Academic Editor for PeerJ